# Stable isotope chemistry reveals plant-dominant diet among early foragers on the Andean Altiplano, 9.0–6.5 cal. ka

**Jennifer C. Chen**[1], **Mark S. Aldenderfer**[2], **Jelmer W. Eerkens**[3], **BrieAnna S. Langlie**[4], **Carlos Viviano Llave**[5], **James T. Watson**[6], **Randall Haas**[7] *

1 Department of Anthropology, The Pennsylvania State University, State College, PA, United States of America, 2 Department of Anthropology and Heritage Studies, University of California, Merced, CA, United States of America, 3 Department of Anthropology, University of California Davis, Davis, CA, United States of America, 4 Department of Anthropology, Binghamton University, Binghamton, NY, United States of America, 5 National Register of Peruvian Archaeologists, Lima, Peru, 6 Arizona State Museum and School of Anthropology, University of Arizona, Tucson, AZ, United States of America, 7 Department of Anthropology, University of Wyoming, Laramie, WY, United States of America

* whaas@uwyo.edu

## Abstract

Current models of early human subsistence economies suggest a focus on large mammal hunting. To evaluate this hypothesis, we examine human bone stable isotope chemistry of 24 individuals from the early Holocene sites of Wilamaya Patjxa (9.0–8.7 cal. ka) and Soro Mik'aya Patjxa (8.0–6.5 cal. ka) located at 3800 meters above sea level on the Andean Altiplano, Peru. Contrary to expectation, Bayesian mixing models based on the isotope chemistry reveal that plants dominated the diet, comprising 70–95% of the average diet. Paleoethnobotanical data further show that tubers may have been the most prominent subsistence resource. These findings update our understanding of earliest forager economies and the pathway to agricultural economies in the Andean highlands. The findings furthermore suggest that the initial subsistence economies of early human populations adapting to new landscapes may have been more plant oriented than current models suggest.

## Introduction

The extent to which early human subsistence economies relied on meat versus plant foods is debated [1, 2]. Current understanding of the earliest subsistence economies of the Andean highlands suggest that meat was the major subsistence resource. Early Holocene assemblages, 11–5 cal. ka, consistently reveal abundant camelid and deer remains, projectile points, and scrapers suggesting hunting-oriented economies [3–11]. For example, Rick [4] concluded that, "The settlement pattern [of the Junín region] and the faunal collections [from the site of Pachamachay] strongly support the hypothesis that vicuña, or similar camelids, were the major food source for *puna* [ecosystem] hunter-gatherers." Such observations are furthermore consistent with diet breadth models, which suggest that early hunter-gatherer populations would tend to target high-ranked large mammals before resorting to plant foods [12–14].

**Data Availability Statement:** All relevant data are within the manuscript and its Supporting Information files.

**Funding:** This research was supported by U.S. National Science Foundation (NSF) grant BCS-1311626 awarded to RH, NSF grant BCS-9816313 awarded to MA, American Philosophical Society Lewis and Clark grant awarded to RH, and funding from the University of California, Davis awarded to RH. The funders had no role in study design, data collection and analysis, decision to publish, or preparation of the manuscript.

**Competing interests:** The authors have declared that no competing interests exist.

Recent analyses of materials from the Archaic Period sites of Soro Mik'aya Patjxa (8.0–6.5 cal. ka) and Wilamaya Patjxa (ca. 8.9 cal. ka), located on the Andean *Altiplano* (High Plateau), provide new opportunities to evaluate these economic models for early highland foragers. Similar to previous research, an abundance of projectile points, scrapers, and lithic debitage indicate considerable investment in the hunting of large terrestrial mammals, likely camelid and deer [15–17]. Preliminary zooarchaeological investigations reveal abundant large-mammal bone, consistent with a hypothesis of large mammal hunting [15, 18]. Groundstone artifacts, though informal and infrequent, suggest some degree of investment in plant resources [15, 18]. Distinctive dental wear patterns on the upper incisors, known as lingual surface attrition of the maxillary anterior teeth suggest intensive tuber processing [19]. Collectively, studies of the Soro Mik'aya Patjxa and Wilamaya Patjxa materials indicate diverse diets of large mammals and plants with an emphasis on large-mammal hunting, consistent with previous findings at other Andean highland sites.

Despite general agreement of various lines of evidence, the evidence remains indirect. Preservation biases that favor projectile points and animal bone compared to plant materials could, to some extent, inflate the hunting signal [20–22]. The biases of previous researchers who have generally been males from a culture in which hunting is a distinctly masculine pursuit could furthermore inflate the hunting signal [23]. It was ostensibly for this reason that ethnography famously revealed plant foods to play a prominent role in forager economies—Arctic economies aside—in contrast to earlier models that emphasized hunting [24–26]. Thus, current archaeological models and evidence leave considerable room for interpretive error.

A more direct but previously unexplored measure of early Andean diets is stable isotope chemistry of human bone. A study of bone isotope chemisty of six early Holocene (8.2–8.0 cal. ka) individuals, including four children and two adults, from the Andean highlands of Argentina finds evidence for tuber and herbivore consumption with breastfeeding and environmental aridity enriching the isotopic values [27]. The current analysis examines the diets of 24 Archaic Period foragers at the highland archaeological sites of Soro Mik'aya Patjxa and Wilamaya Patjxa using stable carbon and nitrogen isotopes in conjunction with more traditional zooarchaeological and paleoethnobotanical approaches. These assemblages date to the early Holocene, collectively spanning 9.0–6.5 cal. ka [16, 18]. Given current models of early highland economies, which point to a mixed diet of animals and plants with an emphasis on large mammals, we should expect to observe the human osteological samples from Soro Mik'aya Patjxa and Wilamaya Patjxa to exhibit dietary carbon ($\delta^{13}C_{diet}$) and nitrogen ($\delta^{15}N_{diet}$) values between those of local fauna and flora with a bias toward the means of the faunal $\delta^{13}C$ and $\delta^{15}N$ values.

This expectation follows from well established relationships between the isotopic composition of human bone and the foods that humans consume [28, 29]. Stable nitrogen isotopes ($\delta^{15}N$) vary with trophic level. Stable carbon isotopes ($\delta^{13}C$) vary with photosynthetic pathway. Such isotopic values in human bone chemistry can provide insights into major subsistence resources including $C_3$ plants, $C_4$ plants, mammals, and fish. Although stable isotope analysis does not offer taxonomic specificity beyond those broad categories, by coupling isotopic insights with zooarchaeological and paleoethnobotanical insights, it may be possible to move beyond preservation biases to gain more accurate estimates of human diets. Given current understanding of highland Archaic diets, we expect zooarchaeological analysis of the Archaic Period sites to reveal an abundance of faunal remains including vicuña, guanaco, or taruca with few small mammal, fish, or bird remains. Paleoethnobotanical analyses should reveal abundant wild chenopod seeds or tuber remains.

## Materials and methods

Recently discovered human burials and other cultural pit features—possibly roasting or storage pits—at the Early-Late Archaic Period archaeological sites of Soro Mik'aya and Wilamaya Patjxa afford an opportunity to evaluate models of early subsistence practices on the Andean Altiplano. A series of radiocarbon dates place Soro Mik'aya Patjxa securely in the Middle to Late Archaic Periods (8.0–6.5 cal. ka) [16]. Radiocarbon dates and artifact typology broadly place Wilamaya Patjxa in the Early to Late Archaic Periods (ca. 11–5 cal. ka), with two direct showing occupation around 9.0 cal. ka [18, 30].

Portions of the two sites were systematically excavated with site matrix and feature fill screened using 6 mm and 1 mm screens, respectively. For each cultural feature, 10-liter bulk soil samples were taken for flotation analysis unless the feature consisted of less than 10L, in which case all feature sediment was collected for flotation. Flotation procedures followed d'Alpoim Guedes et al. [31] and Lennstrom and Hastorf [32] using a modified version of Watson's [33] flotation machine.

The excavations revealed 18 cultural pit features at Soro Mik'aya Patjxa and 39 at Wilamaya Patjxa. From these, 16 individuals were discovered at Soro Mik'aya Patjxa and 12 at Wilamaya Patjxa. Here, we describe the laboratory methods for the three analytical approaches including isotopic, zooarchaeological and paleoethnobotanical approaches.

### Stable isotope analysis of human bone

Human bone samples were excavated and exported under Peruvian Ministry of Culture Permit numbers 064-2013-DGPA-VMPCIC/MC and 138-2015-VMPCIC/MC. Stable isotope chemistry is performed in four different labs including the University of California Davis Stable Isotopes Facility (UCDSIFS), University of California Irvine W.M. Keck Carbon Cycle Accelerator Mass Spectrometer (KCCAMS) Facility, the University of Arizona Accelerator Mass Spectrometry Lab (AMS), and the Penn State University Laboratory for Isotopes and Metals in the Environment (LIME). Collagen extraction for the UCDSIFS and LIME submissions is performed at the UC Davis Archaeometry Lab following the protocol of Eerkens et al. [34]. Collagen extraction for the KCCAMS submissions follows the protocol described by Haas et al. [18].

To assess the extent of diagenetic alterations to bone collagen, we consider atomic C/N ratios with the expectation that reliable readings will exhibit C/N ratios in the range of 3.1–3.6 [35]. As an additional quality control measure, we compare the resultant $\delta^{13}C$ and $\delta^{15}N$ values for three of the individuals—SMP9, SMP16, and WMP6—to values previously reported in radiocarbon analyses performed by The University of Arizona AMS laboratory [16] and the KCCAMS facility [18].

Baseline $\delta^{13}C$ and $\delta^{15}N$ values for candidate subsistence resources, including $C_3$ plants, $C_4$ plants, camelid, or freshwater fish, are compiled from published sources [36–40]. To the extent possible, control samples are restricted to archaeological samples from the central Andean highlands. For any modern samples included, $\delta^{13}C$ values are corrected by +1.5‰ for Suess effects. For any lowland samples included, $\delta^{13}C$ values are offset +2‰ and $\delta^{15}N$ values -1.5‰ based on the regression equations of Szpak et al. [38]. $\delta^{13}C$ bone collagen samples are adjusted using a -2.4‰ offset if terrestrial [41] and -3.7‰ offset if aquatic [42, 43] to adjust for meat-bone offset.

Bayesian mixing models are used to estimate the dietary composition of the Soro Mik'aya Patjxa and Wilamaya Patjxa individuals using the $\delta^{13}C$ and $\delta^{15}N$ values of the consumers and potential food resources. Children ($n = 4$) are excluded from the model to prevent breast-feeding effects from influencing the results. Previous research shows that trophic offsets can vary

widely due to a variety of environmental and trophic effects [27, 44, 45]. Given the antiquity of the system under investigation, the mobility of humans, and the volatility of isoscapes over time, it is difficult to know which trophic correction factors apply to a dietary regime under consideration. We therefore consider the range of possible trophic enrichment factors along with variance terms for both $\delta^{13}$C and $\delta^{15}$N. For $\delta^{13}$C, we use trophic enrichment factors ranging between 4.5–6.0‰, in 0.5‰ increments, with a standard deviation of 0.63‰ [45]. For $\delta^{15}$N, we use trophic enrichment factors ranging between from 3–6‰, in 1‰ increments, with a standard deviation of 0.74‰ [45]. Considering all possible combinations of trophic enrichment factors results in 16 models for evaluation.

All models assume uniformed priors given the lack of prior knowledge on the relative dietary contributions of the broad resource classes. Although it might be tempting to draw on zooarchaeological and archaeobotanical data for informative priors, differential preservation of faunal and floral artifacts precludes this possibility.

Model runs assume both residual and process error, a chain length of 1 million, burn-in of 5000, thinning of 500, and three chains. For dietary estimates, we report median values and 95% credible intervals for each subsistence resource and each model. Model convergence is assessed using Gelmen-Rubin and Geweke diagnostics, and the models are compared to one another using leave-one-out cross-validation information criterion (LOOic) and Akaike Information Criterion (AIC) weights [46]. All computation is performed using R statistical computing language [47] with Bayesian mixing modeling performed using MixSIAR package [46]. Although other packages for mixing modeling are available (e.g., FRUITS [48]), we use MixSIAR [49] because of its currency, documentation, integration with R statistical computing language, and accessibility via open-source Linux operating systems [50]. All code is made available in S1 Data.

## Zooarchaeological analysis

New faunal data are reported for Soro Mik'aya Patjxa. Wilamaya Patjxa data are derived from a previous investigation by Noe [18]. For newly reported materials, all animal bone is weighed and counted. Although abundant, the animal bone is highly fragmented, likely due to intensive processing, making more precise taxonomic identification difficult. The method presented here serves to broadly distinguish between human, small mammal, large mammal, bird, and fish bone. Animal bone fragments are distinguished from human bone based on bone macrostructure where (a) human bone tends to be more porous than animal bone, (b) cortical bone tends to be thicker relative to bone diameter in animals compared to humans, (c) diaphyseal trabecula tends to be present in human but absent in animal bone, and (d) human cranial vaults tend to have thick dipole while animal cranial vaults tend to be more compact [51]. More detailed faunal analysis is ongoing, but the coarse analytical approach taken here is sufficient to address the broad dietary question at hand.

## Paleoethnobotanical analysis

All Soro Mik'aya Patjxa features, which include burial pits and pits of unknown function, are subject to macrobotanical analysis. Samples are sorted using a stereoscopic light microscope with 10 to 40X magnification. Due to environmental conditions in the central Altiplano and the antiquity and exposed nature of the sites, it is highly unlikely that uncarbonized plant remains would have preserved, so analysis is restricted to carbonized remains. Macrobotanical remains are sorted into different tissue categories including seeds, wood, and parenchyma. Parenchyma refers to plant storage tissue. These distinct carbonized tissues with thin-walled cells are believed to be tuber fragments by paleoethnobotanists working in the Altiplano. All

specimens are identified to the most specific taxonomic level possible. Paleoethnobotanical analysis is restricted to Soro Mik'aya Patjxa with Wilamaya Patjxa paleoethnobotanical analysis ongoing.

## Results

Isotopic control samples compiled from the Andean literature include 96 large mammals, 84 $C_3$ plants, 29 $C_4$ plants, and 10 fish samples. The data reveal strong clustering of carbon and nitrogen values by category (Fig 1, Table 1, S1 Table) providing an ideal baseline for comparison with the human bone samples reported here (Table 2). Twenty-four individuals from Soro Mik'aya Patxja (SMP) and Wilamaya Patjxa (WMP) show $\delta^{15}N_{diet}$ values ranging from 2.0‰ to 8.3‰ with a mean of 3.4‰ (see Fig 1A) and $\delta^{13}C_{diet}$ values ranging from -24.3‰ to -22.9‰ with a mean of -23.7‰ (see Fig 1B).

Quality control measures indicate that the archaeological stable isotope values reported here are reliable. Atomic C/N ratios for all but one sample—WMP1—fall within the acceptable range of 3.1–3.6 (see Table 2), indicating that diagenetic processes have not significantly altered the collagen [35]. Furthermore, the $\delta^{13}C$ and $\delta^{15}N$ values from the four previously reported radiocarbon dates [16, 18, 30] are in close agreement with less than 1.3‰ separating values reported among the three labs (Table 3).

The $\delta^{13}C$ and $\delta^{15}N$ values for the human individuals fall closest to the mean of $C_3$ plants with slight enrichment from some other set of resources indicating $C_3$ dominant diets and excluding the possibilities that mammals, $C_4$ plants, or fish comprised large portions of the diets (see Fig 1C). All 16 of the Bayesian mixing models indicate that adult diets were dominated by $C_3$ plants (Table 4). Median $C_3$ plant contribution estimates range from 60–95% and median mammal estimates range from 3–34% for all 16 models. Gelman and Geweke model diagnostics are consistently zero or near zero, indicating that all models are plausible.

The best-fit Bayesian mixing model indicates that $C_3$ plants comprised approximately 84% (73–92%) of the average adult diet with meat comprising just 9% (0–24%), fish 4% (0–13%), and $C_4$ plants 2% (0–6%; Fig 2). This model generated the lowest LOOic value and the highest AIC weight, and is based on a $\delta^{15}N$ trophic enrichment factor of 6.00±0.74‰ and a $\delta^{13}C$ trophic enrichment factor of 5.00±0.63‰. However, five other models produced nearly equivalent AIC weights greater than 0.10 suggesting virtually equivalent model performance. Among these models, median $C_3$ plant estimates range from 70–95% and mammal estimates range between 3–23%. These results show that the particular trophic enrichment factors, ranging from 3.0–6.0‰ for $\delta^{15}N$ and 4.5–6.0 for $\delta^{13}C$, have little effect on the broad dietary estimates. All models indicate a plant dominant diet with median values for $C_3$ plants ranging between 70–95% and mammals ranging between 3–23%. Thus all credible subsistence models indicate that plant foods comprised the majority of individual diets and meat played a secondary role. These findings are inconsistent with the working hypothesis of a meat-dominant diet and instead suggest a plant-dominant diet among early forager populations of the Andean Altiplano, 9.0–6.5 cal. ka.

Zooarchaeological and paleoethnobotanical observations offer some taxonomic precision beyond the broad food categories used in the stable isotope analysis. Excavations at Soro Mik'aya produced 3193 fragments of animal bone (number of individual specimens or NISP) from across the site. Of the total specimens, the most frequently identified category included 200 large mammal fragments with only trace amounts of small mammal, bird, and fish bones reported [15]. Most of these large mammal bones were burned (n = 150; 75%). A previously published assessment of 341 faunal bone elements from Wilamaya Patjxa faunal assemblage similarly revealed that camelid and deer bone were the most frequently identified taxa with 17

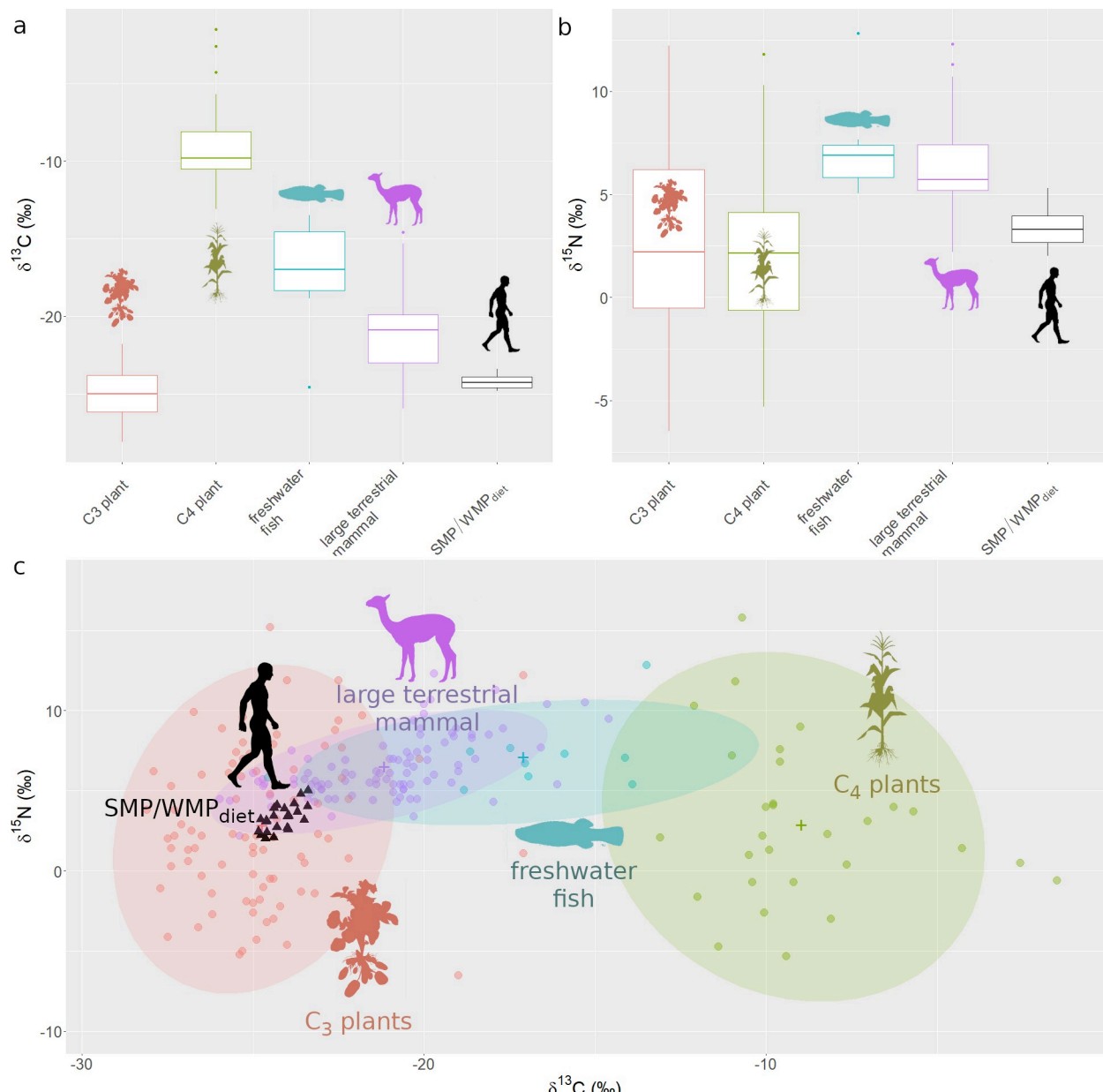

**Fig 1. Carbon and nitrogen plots for control samples and 25 human bone samples from Soro Mik'aya Patjxa and Wilamaya Patjxa, indicating a plant-dominant diet.** a) $\delta^{13}C_{diet}$ values are consistent with those of $C_3$ plants with slight enrichment from some other resource types. b) $\delta^{15}N_{diet}$ values are most consistent with those of plants. c) Biplot of $\delta^{13}C$ and $\delta^{15}N$ values are consistent with a mixed diet principally based on $C_3$ plants with low levels of enrichment from some other resource. Dietary values assume $\delta^{13}C$ TEF = 5.0‰, $\delta^{15}N$ TEF = 6.0‰ based on mixing model results (see Table 4). Dots = individual samples, ellipses = 95% variance ranges for each category, and crosshairs = mean values by category (see Tables 1 and 2).

camelid and 5 deer elements observed [18]. Notably absent, again, are small mammals, birds, and fish. These data indicate that the slight enrichment observed in the carbon and nitrogen values in the human bone was likely due to large mammal consumption and not small mammals, fish, or birds.

The most abundant plant-food specimens in the Soro Mik'aya Patjxa paleoethnobotanical assemblage is parenchyma tissue, identified as tuber fragments with 52 specimens found in 9

**Table 1. Summary statistics for stable isotopic control data for high-altitude Andean food resources and archaeological samples from Soro Mik'aya Patjxa (SMP) and Wilamaya Patjxa (WMP).** See S1 Table for sample data.

| population | n | $\delta^{13}C$ mean (‰) | $\delta^{13}C$ sd (‰) | $\delta^{15}N$ mean (‰) | $\delta^{15}N$ sd (‰) | references |
|---|---|---|---|---|---|---|
| $C_3$ plants | 84 | -24.7 | 2.1 | 2.9 | 4.6 | [36, 40] |
| $C_4$ plants | 29 | -9.0 | 2.7 | 2.9 | 4.9 | [36, 38, 40] |
| freshwater fish | 10 | -17.1 | 3.2 | 7.1 | 2.2 | [40] |
| large terrestrial mammal | 96 | -21.2 | 2.3 | 6.5 | 2.4 | [39, 52–54] |
| SMP/WMP$_{diet}$* | 24 | -23.7 | 0.4 | 3.4* | 0.9 | NA |

*Dietary values assume $\delta^{13}C$ TEF = 5.0‰, $\delta^{15}N$ TEF = 6.0‰ based on mixing model results (see Table 4).

of 15 features (Fig 3, Table 5). Chenopod seeds are nearly absent with just three wild specimens observed among three features. The most abundant paleoethnobotanical samples are non-food resources including wood fragments (*n* = 448) and grass (*Poaceae*) seeds (*n* = 161) observed in

**Table 2. Human bone collagen isotopic results for Soro Mik'aya Patjxa and Wilamaya Patjxa individuals.** See S2 Table for additional metadata.

| burial | age class[a] | element | lab[b] | $\delta^{13}C_{raw}$ (‰) | $\delta^{15}N_{raw}$ (‰) | $\delta^{13}C_{diet}$[c] (‰) | $\delta^{15}N_{diet}$[c] (‰) | atomic C/N | date (95% cal. BP)[d] |
|---|---|---|---|---|---|---|---|---|---|
| SMP 1 | child | parietal (squama) | UCDSIF | -18.1 | 10.8 | -23.1 | 4.8 | 3.2 | n.d. |
| SMP 2 | adult | temporal (petrous portion) | UCDSIF | -18.9 | 9.9 | -23.9 | 3.9 | 3.3 | n.d. |
| SMP 3 | adult | rib 10 (left) | UCDSIF | -18.9 | 9.2 | -23.9 | 3.2 | 3.2 | 7565–7177 [16] |
| SMP 4 | adolescent | rib 1 (left) | UCDSIF | -19.3 | 8.2 | -24.3 | 2.2 | 3.2 | 7565–7177 [16] |
| SMP 5 | adult | hand mid phalanx (right) | UCDSIF | -18.7 | 11.3 | -23.7 | 5.3 | 3.2 | 6856–6569 [16] |
| SMP 6 | adult | metatarsal 5 (left) | UCDSIF | -19.3 | 9.2 | -24.3 | 3.2 | 3.3 | 7153–6756 [16] |
| SMP 7 | adolescent | hand proximal phalanx (right) | UCDSIF | -19.1 | 8 | -24.1 | 2 | 3.2 | 6780–6510 [16] |
| SMP 8 | adult | hand proximal phalanx (right) | UCDSIF | -19.3 | 8.5 | -24.3 | 2.5 | 3.2 | 7160–6885 [16] |
| SMP 9 | adult | hand proximal phalanx (left) | UCDSIF | -18.5 | 8.6 | -23.5 | 2.6 | 3.2 | 7465–7317 [16] |
| SMP 10 | adult | hand proximal phalanx (left) | UCDSIF | -18.2 | 9.7 | -23.2 | 3.7 | 3.2 | 6907–6574 [16] |
| SMP 11 | adult | rib (right) | UCDSIF | -18.9 | 9.4 | -23.9 | 3.4 | 3.2 | 6883–6669 [16] |
| SMP 12 | adult | rib (left) | UCDSIF | -19.1 | 9.1 | -24.1 | 3.1 | 3.2 | n.d. |
| SMP 13 | child | temporal (petrous portion) | UCDSIF | -17.9 | 11 | -22.9 | 5 | 3.3 | 6883–6669 [16] |
| SMP 15 | adult | hand proximal phalanx (left) | UCDSIF | -18.8 | 8.7 | -23.8 | 2.7 | 3.2 | n.d. |
| SMP 16 | adult | mandible (right) | UCDSIF | -18.8 | 10.1 | -23.8 | 4.1 | 3.3 | 7247–7009 [16] |
| WMP 1 | adult | tibia frag (side indeterminate) | UCDSIF | -18.5 | 9.5 | -23.5 | 3.5 | 3.9 | 8990–8650 [30] |
| WMP 2 | adult | long bone diaphysis fragment | KCCAMS | -18.6 | 8.7 | -23.6 | 2.7 | 3.3 | n.d. |
| WMP 3 | adult | bone frag | UCDSIF | -18.6 | 9.9 | -23.6 | 3.9 | 3.4 | n.d. |
| WMP 5 | adult | left petrous portion | UCDSIF | -17.9 | 10 | -22.9 | 4 | 3.4 | n.d. |
| WMP 6 | adolescent | indeterminate bone fragment | UCDSIF | -19.1 | 9.2 | -24.1 | 3.2 | 3.6 | 8992–8651 [18] |
| WMP 7 | adult | left scapula | KCCAMS | -19.1 | 8.4 | -24.1 | 2.4 | 3.2 | n.d. |
| WMP 8 | adolescent | 3$^{rd}$ molar | LIME | -18.7 | 9.8 | -23.7 | 3.8 | 3.2 | n.d. |
| WMP 9 | child | cranial fragment | KCCAMS | -18.5 | 8.7 | -23.5 | 2.7 | 3.2 | n.d. |
| WMP 10 | child | cranial fragment | KCCAMS | -18.3 | 10.2 | -23.3 | 4.2 | 3.2 | n.d. |
| | | mean | | -18.7 | 9.4 | -23.7 | 3.4 | | |
| | | standard deviation | | 0.4 | 0.9 | 0.4 | 0.9 | | |

*age classes as defined by Buikstra and Ubelaker [55]

[b]UCDSIF = UC Davis Stable Isotope Facility; KCCAMS = UC Irvine Keck Carbon Cycle Accelerator Mass Spectrometry facility; LIME = Penn State University Laboratory for Isotopes and Metals in the Environment

[c]Dietary values assume $\delta^{13}C$ TEF = 5.0‰, $\delta^{15}N$ TEF = 6.0‰ based on mixing model results (see Table 4).

[d]calibrated using Southern Hemisphere Calibration Curve 2020 [56].

**Table 3. Inter-laboratory comparison of isotopic results showing consistent results and acceptable atomic C/N ratios.** UCDSIFS compared to AMS and KCCAMS.

| individual | δ13C (‰) | δ15N (‰) | 14C comparisons | | | | | |
|---|---|---|---|---|---|---|---|---|
| | | | lab | lab ID | 14C age (B.P.) | C/N$_{atomic}$ | δ13C (‰) | δ15N (‰) |
| SMP 9 | -18.5 | 8.6 | AMS | AA107345 [16] | 6529±41 | 3.2 | -19.2 | 7.9 |
| SMP 16 | -18.8 | 10.1 | AMS | AA107490 [16] | 6259±38 | 3.3 | -18.8 | 9.7 |
| WMP 1 | -18.5 | 9.5 | KCCAMS | UCIAMS 259854 [30] | 8010±25 | 3.3 | -18.4 | 8.6 |
| WMP 6 | -19.11 | 9.2 | KCCAMS | UCIAMS 212748 [18] | 8035±20 | 3.2 | -18.8 | 8.2 |
| | | | KCCAMS | UCIAMS 212749 [18] | 7965±25 | 3.3 | -19.0 | 8.0 |

nearly every feature. The wood most likely reflects use as fuel. Similarly the small grass seeds likely reflect fuel use whether from burning grasses or the dung of camelids [57]. The small grass seeds are typical of the region, and none of the taxa are suitable for human consumption but are excellent forage for camelids. This finding is consistent with a model of early wild tuber consumption and other lines of evidence that suggest intensive tuber use in the region during the Archaic Period [19, 58, 59].

## Discussion

The stable isotope, faunal, and paleoethnobotanical evidence from the sites of Soro Mik'aya Patjxa and Wilamaya Patjxa converge to indicate that C$_3$ plants, likely wild tubers, comprised the major component of early forager diets on the Andean Altiplano and that meat, including

**Table 4. Bayesian mixing model comparison considering different trophic enrichment factors.**

| model | TEF[a] | | estimated dietary contribution (%)[b] | | | | Gelman diagnostic[c] | Geweke diagnostic[d] | LOOic[e] | Akaike weight[f] |
|---|---|---|---|---|---|---|---|---|---|---|
| | δ13C | δ15N | C$_3$ plants | C$_4$ plants | fish | mammals | | | | |
| 1 | 4.5 | 3.0 | 60(42–76) | 0(0–3) | 4(0–47) | 34(2–52) | 0,0,0 | 0,0,1 | -38.5 | 0.00 |
| 2 | 5.0 | 3.0 | 75(59–88) | 1(0–3) | 4(0–22) | 19(1–36) | 2,0,0 | 0,0,1 | -36.3 | 0.00 |
| 3 | 5.5 | 3.0 | 86(75–95) | 0(0–3) | 3(0–12) | 9(0–22) | 1,0,0 | 0,3,2 | -34.9 | 0.00 |
| 4 | 6.0 | 3.0 | 94(86–98) | 0(0–2) | 1(0–6) | 4(0–12) | 1,0,0 | 1,2,2 | -35.5 | 0.00 |
| 5 | 4.5 | 4.0 | 62(50–78) | 1(0–3) | 4(0–21) | 32(6–48) | 0,0,0 | 0,0,1 | -49.8 | 0.03 |
| 6 | 5.0 | 4.0 | 75(63–87) | 1(0–3) | 4(0–16) | 20(2–36) | 0,0,0 | 0,0,1 | -48.7 | 0.02 |
| 7 | 5.5 | 4.0 | 86(75–95) | 1(0–3) | 3(0–11) | 9(0–23) | 0,0,0 | 0,6,0 | -48.0 | 0.01 |
| 8 | 6.0 | 4.0 | 94(85–98) | 0(0–2) | 1(0–6) | 4(0–13) | 0,0,0 | 1,6,0 | -47.6 | 0.01 |
| 9 | 4.5 | 5.0 | 70(57–83) | 1(0–5) | 5(0–19) | 23(3–40) | 0,0,0 | 1,9,0 | -52.6 | 0.13 |
| 10 | 5.0 | 5.0 | 78(67–89) | 1(0–4) | 4(0–14) | 16(1–31) | 0,0,0 | 1,5,1 | -52.2 | 0.11 |
| 11 | 5.5 | 5.0 | 88(77–95) | 1(0–3) | 3(0–10) | 8(0–21) | 0,0,0 | 0,1,2 | -51.7 | 0.09 |
| 12 | 6.0 | 5.0 | 94(86–98) | 0(0–2) | 1(0–6) | 3(0–12) | 0,0,0 | 1,3,0 | -51.1 | 0.06 |
| 13 | 4.5 | 6.0 | 79(67–88) | 3(0–8) | 6(0–17) | 10(1–28) | 0,0,0 | 1,10,0 | -52.6 | 0.13 |
| *14 | 5.0 | 6.0 | 84(73–92) | 2(0–6) | 4(0–13) | 9(0–24) | 0,0,0 | 2,0,0 | -52.7 | 0.14 |
| 15 | 5.5 | 6.0 | 90(81–96) | 1(0–4) | 2(0–9) | 6(0–17) | 0,0,0 | 0,9,1 | -52.6 | 0.13 |
| 16 | 6.0 | 6.0 | 95(88–98) | 0(0–2) | 1(0–6) | 3(0–10) | 0,0,0 | 0,0,9 | -52.4 | 0.12 |

[a]TEF = trophic enrichment factor

[b]posterior probability median (95% range)

[c]variables > 1.01, variables > 1.05, variables > 1.1 (27 variables)

[d]number of variables outside 95% confidence level for each of three chains (should be <2, or 5% of 27 variables)

[e]leave-one-out cross validation information criterion (LOOic) for assessing model efficacy. Smaller values indicate more powerful models.

[f]Akaike weight for model selection. Larger values indicate more powerful models.

*best approximating model based on LOOic and Akaike weight.

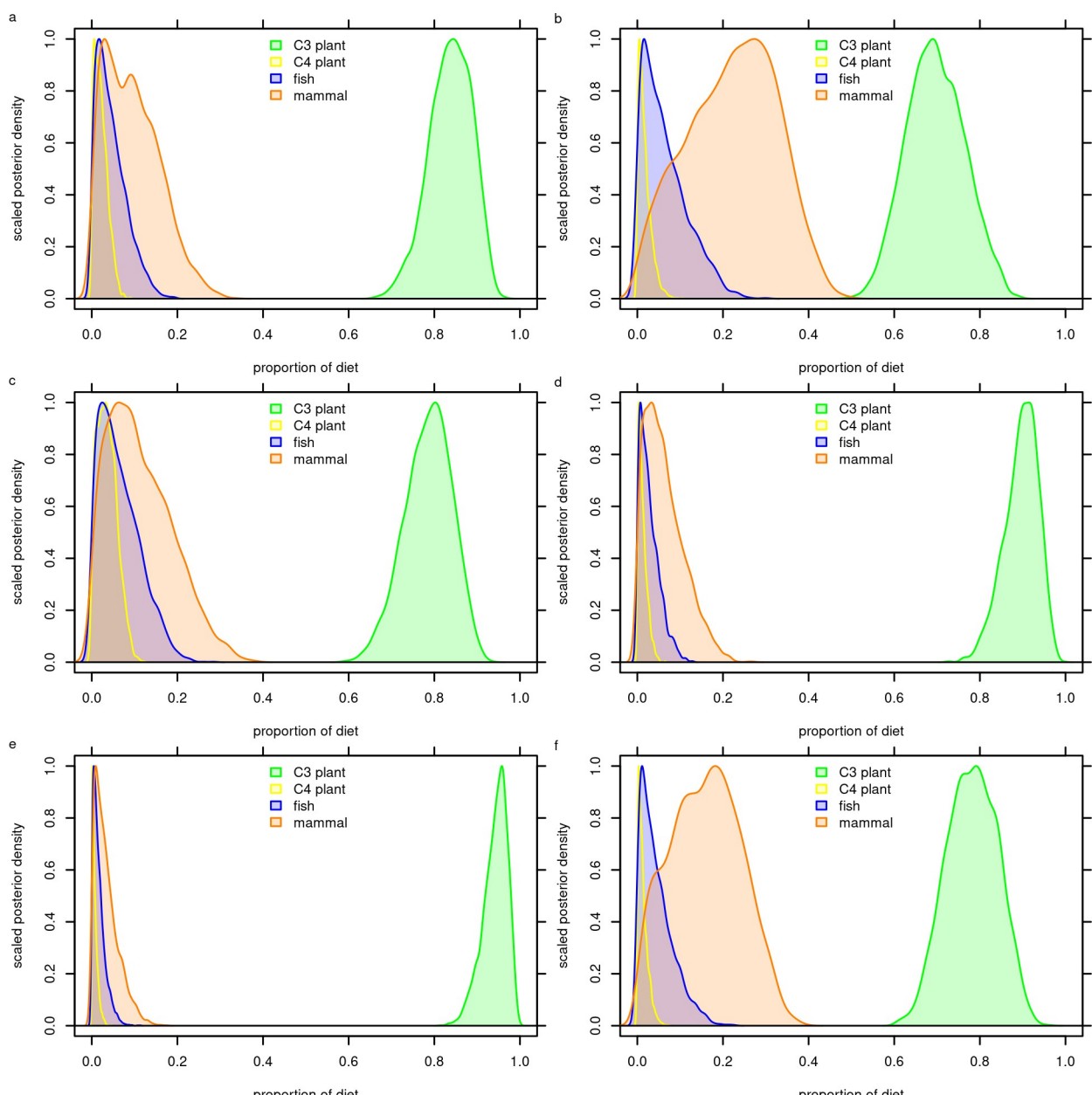

**Fig 2. Bayesian mixing model results for the six best-fit models showing that C$_3$ plants comprised the majority of the diet and mammals played a secondary role in the subsistence economies of Soro Mik'aya Patjxa and Wilamaya Patjxa.** a. model 14, b. model 9, c. model 13, d. model 15, e. model 16, f. model 10 (see Table 4).

vicuña and taruca, played a secondary role. C$_4$ plants, small mammals, fish, and birds appear to have played negligible roles in these early subsistence economies. The findings presented here depart from current thinking about early Andean highland diets and force a reconsideration of existing economic models.

One possible explanation for the unexpected emphasis on plant foods among this early highland population may be that large mammal populations had been severely reduced by 9 cal. ka. Although the Wilamaya Patjxa assemblage includes the earliest archaeological period of the region—the Early Archaic Period, 11.0–9.0 cal. ka—it is restricted to the latter end of the

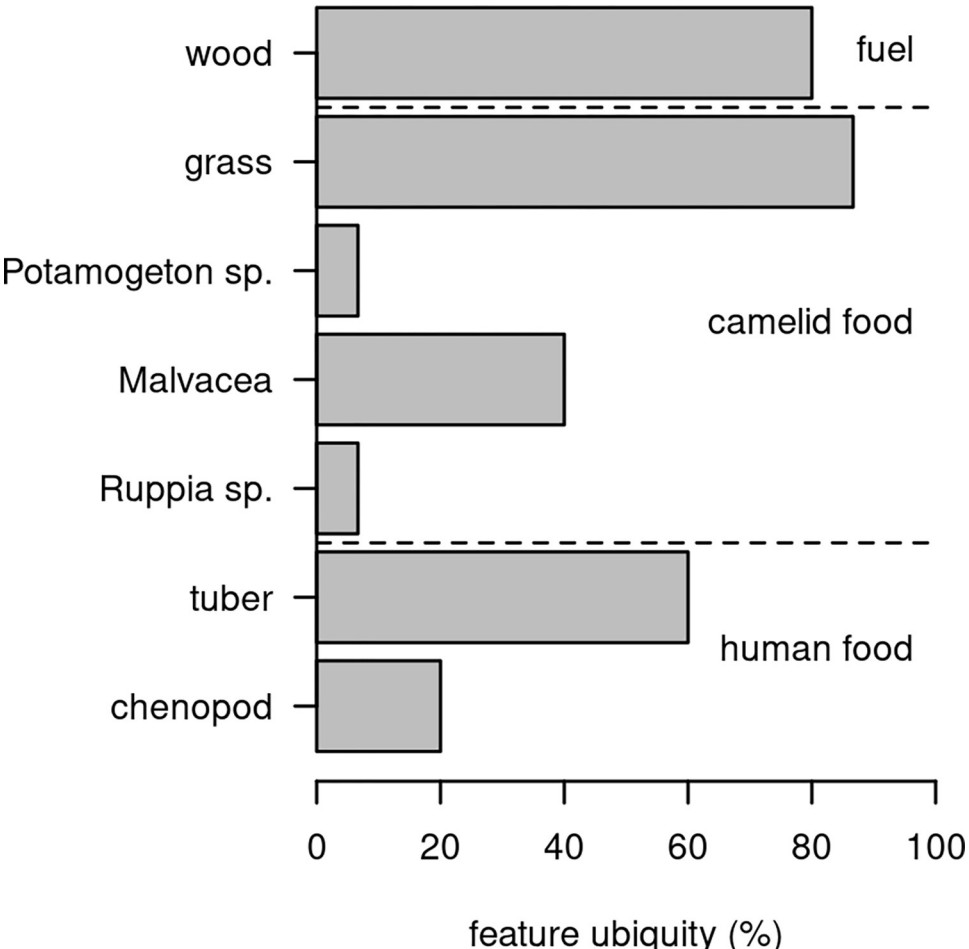

**Fig 3. Results of paleoethnobotanical analysis from features at Soro Mik'aya Patjxa.** Ubiquity is measured as the proportion of archaeological features containing each taxon. Values based on 15 features, 300L of flotation, and 688 paleoethnobotanical artifacts (see Table 5).

period [18]. It is currently unclear when human populations first arrived in the region [60], but if they arrived as early as 11 cal. ka., then humans would have been hunting the region for 2,000 years prior to the earliest individuals under investigation. This would certainly have been enough time to decimate the region's animal population in the absence of animal conservation strategies [4]. A second possibility is that the earliest populations simply did not engage in hunting to the extent previously thought. Previous research suggests that prey choice models may over-estimate the dietary values of large mammals whether due to physical risk [61], prey behavior [62], or economic risks associated with long encounter intervals [63]. A third explanatory possibility is that early highland populations relied heavily on hunting large mammals but incorporated animal digesta into their diets [64], which could simultaneously account for both the archaeological signatures of hunting—animal bone and projectile points—and the depleted nitrogen values in human bone chemistry observed here. Evaluating these hypotheses will ultimately require investigation of earlier archaeological assemblages.

These findings furthermore hold implications for our understanding of domestication in the high Andes. Pearsall [65] and Kuznar [66] proposed that plant management commenced after camelids were managed in corrals. In this view, camelids transported plant seeds to corrals where they thrived in soils fertilized by camelid dung, creating a mutually beneficial

**Table 5. Carbonized macrobotanical materials from Soro Mik'aya Patjxa flotation samples.**

| feature | flotation volume (L) | human food | | camelid food | | | | fuel | total |
|---|---|---|---|---|---|---|---|---|---|
| | | *Chenopodium* sp. | parenchyma | *Ruppia* sp. | Malvaceae | *Potamogeton* sp. | *Poaceae* | wood | |
| 2 | 27 | 0 | 25 | 0 | 2 | 0 | 3 | 57 | 87 |
| 3 | 23 | 0 | 1 | 1 | 0 | 0 | 8 | 20 | 30 |
| 4 | 31 | 1 | 4 | 0 | 12 | 0 | 57 | 75 | 149 |
| 5 | 25 | 0 | 8 | 0 | 2 | 0 | 29 | 33 | 72 |
| 6 | 18 | 0 | 1 | 0 | 1 | 1 | 4 | 35 | 42 |
| 7 | 1 | 0 | 0 | 0 | 0 | 0 | 1 | 0 | 1 |
| 8 | 5 | 0 | 0 | 0 | 0 | 0 | 0 | 2 | 2 |
| 9 | 10 | 0 | 3 | 0 | 0 | 0 | 5 | 7 | 15 |
| 10 | 25 | 1 | 0 | 0 | 2 | 0 | 10 | 91 | 104 |
| 13 | 33 | 0 | 1 | 0 | 0 | 0 | 2 | 19 | 22 |
| 14 | 12 | 0 | 6 | 0 | 0 | 0 | 1 | 49 | 56 |
| 14/15* | 13 | 0 | 2 | 0 | 0 | 0 | 4 | 41 | 47 |
| 15 | 10 | 0 | 0 | 0 | 0 | 0 | 6 | 8 | 14 |
| 16 | 48 | 1 | 1 | 0 | 3 | 0 | 28 | 0 | 33 |
| 17 | 9 | 0 | 0 | 0 | 0 | 0 | 0 | 0 | 0 |
| 18 | 10 | 0 | 0 | 0 | 0 | 0 | 3 | 11 | 14 |
| total | 300 | 3 | 52 | 1 | 22 | 1 | 161 | 448 | 688 |

*sample contexts mixed. Excluded from ubiquity calculations.

relationship between camelids, plants, and human communities. Versions of this model suggest that *Chenopodium* spp. (including the crops quinoa and kañawa) and the tuber maca (*Lepidium mevenii* Walp.) were domesticated in this way. These coevolutionary processes also likely led to the domestication of potatoes (*Solanum tuberosum* L.) and up to 15 other species of roots and tubers in the Andes [67]. Consistent with this model, chenopod seeds, maca tubers, and managed camelids appeared after about 4,000 years ago at the rock shelter site of Panalauca where the sizes of chenopod and maca specimens increased through time [65]. Recent research into domestication mutualism supports the Andean version of the camp follower hypothesis showing that chenopods, tubers, and camelids were likely domesticated in tandem as complementary foods [68].

While current models suggest initial co-evolutionary processes involving maca, chenopod, and camelid intensification in the highlands, the paleoethnobotanical evidence presented here fails to find strong evidence of early intensive chenopod use on the Altiplano during the Archaic period. This may reflect preservation biases given the small and delicate nature of wild chenopod seeds. Alternatively, it may be that chenopods did not become economically important in the region until sometime after 6.5 cal. ka. A later incorporation of chenopods into the diet would be consistent with prey choice models given the low post-encounter return rates of small seeds relative to tubers [69].

The finding of tuber fragments at Soro Mik'aya Patjxa is consistent with the role of tubers in the early stages of the co-evolutionary process. However, the tuber fragments are unlikely to be maca, which is a more northerly taxon. Several tuber species could potentially account for the tubers observed at Soro Mik'aya Patjxa, but they are most likely associated with wild potato species, which are concentrated in the region and were likely domesticated there [59, 70–72]. Additional sites on the Altiplano should be examined with particular attention to contemporaneous sites that would assess replicability of the current findings and to non-contemporaneous sites that would afford diachronic comparisons.

In sum, the results presented here are consistent with a model of human-tuber-camelid co-evolutionary dynamics beginning approximately 9,000 years ago on the Andean Altiplano. These findings support an updated model of Archaic Period subsistence practices in the central high Andes in which forager subsistence economies 9.0–6.5 cal. ka. emphasized plant foraging with lesser attention to large mammal hunting and a virtual absence of small animal hunting and fishing. This resource base may have catalyzed potato and camelid domestication in the subsequent Terminal Archaic Period after 5 cal. ka [58].

These findings further highlight the need to re-evaluate anthropological understanding of early forager diets more generally. Current perspectives vary with some models emphasizing the primacy of plants and others of animals [1, 2] with plant foraging becoming increasingly important relatively late in time on the eve of agriculture [73]. This may still be so, but the current analysis suggests that the shift to plant-foraging economies may have happened relatively rapidly, evidently having transpired in less than 2,000 years in the Andean case. This observation resonates with recent archaeological theory and findings that reveal a prominent role for plant foods in early forager diets [31, 74, 75] and ethnographic findings of the 1960s when, contrary to dominant thinking of the time, many subsistence economies once thought to be meat-dominant were shown to be plant-dominant [24, 25]. Stable isotope chemistry gives archaeologists the opportunity to reliably extend such investigations into the deep past. The current study arrives at a similar place to the earlier ethnographic findings—plant foods were central to early human economies.

## Supporting information

**S1 Data. Bayesian mixing model code in R language.**
(R)

**S1 Table. Isotopic control data.**
(CSV)

**S2 Table. Isotopic data for Soro Mik'aya Patjxa and Wilamaya Patjxa individuals.**
(CSV)

## Acknowledgments

Field support was provided by Collasuyo Archaeological Research Institute, and the communities of Mulla Fasiri and Totorani, Peru. Bryna Hull (UC Davis) provided support for stable isotope analysis. Samples of human remains were excavated and exported under Peruvian Ministry of Culture Permit numbers 064-2013-DGPA-VMPCIC/MC and 138-2015-VMPCIC/MC.

## Author Contributions

**Conceptualization:** Jennifer C. Chen, Randall Haas.

**Data curation:** Jennifer C. Chen, Mark S. Aldenderfer, Carlos Viviano Llave, Randall Haas.

**Formal analysis:** Jennifer C. Chen, BrieAnna S. Langlie, James T. Watson, Randall Haas.

**Funding acquisition:** Mark S. Aldenderfer, Randall Haas.

**Investigation:** Jennifer C. Chen, Mark S. Aldenderfer, BrieAnna S. Langlie, Carlos Viviano Llave, James T. Watson, Randall Haas.

**Methodology:** Jennifer C. Chen, Jelmer W. Eerkens, BrieAnna S. Langlie, James T. Watson, Randall Haas.

**Project administration:** Carlos Viviano Llave, Randall Haas.

**Resources:** Mark S. Aldenderfer, Jelmer W. Eerkens, BrieAnna S. Langlie, Randall Haas.

**Supervision:** Jelmer W. Eerkens, Randall Haas.

**Validation:** Jennifer C. Chen, Jelmer W. Eerkens, Randall Haas.

**Visualization:** Jennifer C. Chen, Randall Haas.

**Writing – original draft:** Jennifer C. Chen, Randall Haas.

**Writing – review & editing:** Jennifer C. Chen, Mark S. Aldenderfer, Jelmer W. Eerkens, BrieAnna S. Langlie, Carlos Viviano Llave, James T. Watson, Randall Haas.

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
