## [Decision Letter · Decision Letter 0]

12 Sep 2023

PONE-D-23-22494Stable isotope chemistry of human bone reveals plant-dominant diet among early foragers on the Andean Altiplano, 9.0–6.5 cal. kaPLOS ONE

Dear Dr. Haas,

Thank you for submitting your manuscript to PLOS ONE. After careful consideration, we feel that it has merit but does not fully meet PLOS ONE’s publication criteria as it currently stands. Therefore, we invite you to submit a revised version of the manuscript that addresses the points raised during the review process.

The manuscript by Chen and colleagues addresses the subsistence economy of early Holocene foragers from the central Andean highlands by using stable isotope analysis in the context of previous hypotheses about a diet dominated by animal resources. Three reviewers evaluated the manuscript and despite the topic being appropriate for the journal, and I believe this can be an important contribution, the reviewers also highlight a series of limitations in the current version of the manuscript including the quality of the stable isotope data investigated, the offsets and enrichment factors used, misinterpretations of the evidence recovered and interpretative errors. I agree with the suggestions made by the reviewers, and I recommend the authors address them by putting effort into clarifying the points mentioned. Please take into account that at least one reviewer suggested the rejection of this manuscript in its current version.

Besides the suggestions by the reviewers, I recommend the authors correct the following issues:

1). The manuscript organization does not match the submission guidelines

2). You need to include more details regarding the sample composition as well as add the estimated sex and age of the individuals investigated.  Additional analyses comparing sex and age groups would add further information. 

3). The isotopic methodology used and its application in archaeology must be expanded

4). More details on the Bayesian mixing models are also necessary. Since it is possible to use prior information would be important to model the human diet composition by evaluating the differential contribution of C3, C4 plants and animals feeding on such resources.    

5). In general, the materials and methods are poorly described including the dataset investigated.

6). The high nitrogen values found among some adults/adolescents ≥ 10‰ are interesting. In some cases, in the Andes people used animal-derived fertilizers which could increase the nitrogen values, have you evaluated such a possibility?

7). In the discussion, what do you mean by geographic evidence?

We look forward to receiving your revised manuscript.

Kind regards,

Miguel Delgado, PhD

División Antropología, Facultad de Ciencias Naturales y Museo, Universidad Nacional de La Plata (Argentina)

Academic Editor

PLOS ONE

5. We note that you have referenced **(Samantha M. Michell Animals of the cloud forest: isotopic variation of archaeological faunal remains from Kuelap, Peru. Unpublished Ph.D. dissertation. University of Central Florida, Orlando. (2014))** and **(Watson JT, McClelland J. Distinguishing human from non-human animal bone. Unpublished manuscript of the Arizona State Museum. Tucson 2018)** which have currently not yet been accepted for publication. Please remove this from your References and amend this to state in the body of your manuscript: (ie “Bewick et al. [Unpublished]”) as detailed online in our guide for authors

6. We note that Figure 1 in your submission contain copyrighted images. All PLOS content is published under the Creative Commons Attribution License (CC BY 4.0), which means that the manuscript, images, and Supporting Information files will be freely available online, and any third party is permitted to access, download, copy, distribute, and use these materials in any way, even commercially, with proper attribution. For more information, see our copyright guidelines: http://journals.plos.org/plosone/s/licenses-and-copyright.

Additional Editor Comments:

The manuscript by Chen and colleagues addresses the subsistence economy of early Holocene foragers from the central Andean highlands by using stable isotope analysis in the context of previous hypotheses about a diet dominated by animal resources. Three reviewers evaluated the manuscript and despite the topic being appropriate for the journal, and I believe this can be an important contribution, the reviewers also highlight a series of limitations in the current version of the manuscript including the quality of the stable isotope data investigated, the offsets and enrichment factors used, misinterpretations of the evidence recovered and interpretative errors. I agree with the suggestions made by the reviewers, and I recommend the authors address them by putting effort into clarifying the points mentioned. Please take into account that at least one reviewer suggested the rejection of this manuscript in its current version.

Besides the suggestions by the reviewers, I recommend the authors correct the following issues:

1). The manuscript organization does not match the submission guidelines.

2). You need to include more details regarding the sample composition as well as add the estimated sex and age of the individuals investigated. Additional analyses comparing sex and age groups would add further information.

3). The isotopic methodology used and its application in archaeology must be expanded.

4). More details on the Bayesian mixing models are also necessary. Since it is possible to use prior information would be important to model the human diet composition by evaluating the differential contribution of C3, C4 plants and animals feeding on such resources.

5). In general, the materials and methods are poorly described including the dataset investigated.

6). The high nitrogen values found among some adults/adolescents ≥ 10‰ are interesting. In some cases, in the Andes people used animal-derived fertilizers which could increase the nitrogen values, have you evaluated such a possibility?

7). In the discussion, what do you mean by geographic evidence?

Reviewers' comments:

Reviewer's Responses to Questions

**Comments to the Author**

1. Is the manuscript technically sound, and do the data support the conclusions?

Reviewer #1: Partly

Reviewer #2: Partly

Reviewer #3: Partly

2. Has the statistical analysis been performed appropriately and rigorously? 

Reviewer #1: No

Reviewer #2: Yes

Reviewer #3: I Don't Know

3. Have the authors made all data underlying the findings in their manuscript fully available?

Reviewer #1: No

Reviewer #2: Yes

Reviewer #3: No

4. Is the manuscript presented in an intelligible fashion and written in standard English?

Reviewer #1: Yes

Reviewer #2: Yes

Reviewer #3: Yes

5. Review Comments to the Author

Reviewer #1: The manuscript shows the analysis of an impressive human skeletal sample, both for the structure of the sample and for its chronology. However, the authors do not show sufficient rigor in the analysis.

First, the authors omit basic information. Without it, it is impossible to evaluate a manuscript on stable isotopes: C/N of each sample, %C, %N and recovered collagen yield. It is also relevant to include the internal standards used in the laboratories.

Second, the terrestrial fauna values used in their model are not appropriate. Well, they correspond to animals associated with late Holocene chronologies, even associated with agropastoral contexts. Much more appropriate are the values exposed in López et al. 2017, Mondini & Panarello 2014, and Grant et al. 2018. Also, the authors do not discuss values generated for contemporary and Andean cases, as is the case exposed in Killian Galván et al. 2016.

The information about the isotopic composition of the resources is poorly presented: the supplementary table does not indicate the scientific or common names of all the species and does not include the isotopic information of the archaeofauna and botanical specimens found in the archaeological site.

Lastly, the conclusion reached by the authors is disruptive but is not supported by the information they present. Half of the individuals analyzed show diets based on the consumption of terrestrial animals. This is observed with the coincidences between the estimated diets and the ellipse corresponding to that resource (figure 1).

In summary, the low values (not "enriched", please do not misuse this term), both carbon and nitrogen, could be perfectly explained by the consumption of tubers. But for this, we must first ensure that the values are primary and that the resources to infer the diet are credible.

Suggested bibliography for authors

-López M, P., Cartajena, I., Loyola, R., Núñez, L., & Carrasco, C. (2017). The use of hunting and herding spaces: Stable isotope analysis of Late Archaic and Early Formative camelids in the Tulan transect (Puna de Atacama, Chile). International Journal of Osteoarchaeology, 27(6), 1059-1069.

- Mondini, M., & Panarello, H. (2014). Isotopic evidence in Holocene camelids from the southern Puna. In Libro de Resúmenes, 12th International Conference of the International Council for ArchaeoZoology (Vol. 114).

- Grant, J., Mondini, M., & Panarello, H. O. (2018). Carbon and nitrogen isotopic ecology of Holocene camelids in the Southern Puna (Antofagasta de la Sierra, Catamarca, Argentina): Archaeological and environmental implications. Journal of Archaeological Science: Reports, 18, 637-647.

- Killian Galván, V., Martínez, J., Cherkinsky, A., Mondini, M., & Panarello, H. (2016). Stable isotope analysis on human remains from the final Early Holocene in the southern Puna of Argentina: the case of Peñas de las Trampas 1.1. Environmental Archaeology, 21(1), 1-10.

Suggested bibliography for authors

Reviewer #2: The paper “Stable isotope chemistry of human bone reveals plant-dominant diet among early foragers on the Andean Altiplano, 9.0–6.5 cal. Ka” by Jennifer Chen et al. presents the results of stable isotope data from 23 individuals from the early Holocene sites of Wilamaya Patjxa and Soro Mik’aya Patjxa in Peru. This is a well-written paper that presents new data on an important period within South American archaeology and it has implications for improving the diversity of hunter-gatherer lifeways and for understanding the ecological context in which plants and animals were domesticated. Although the study is well-designed and the data are trustworthy, I have some concerns about the main argument and interpretation of the data presented in the paper. Until these concerns are addressed, I recommend accept with major revisions.

The most serious issue I have with the data interpretation concerns the nitrogen trophic enrichment factor (TEF) used by the authors, which they set at 6 per mil. Within a food web, an enrichment of 15N relative to 14N in proteinaceous tissues occurs between trophic positions. This is most often cited as being about 3-5 per mil, with carnivores being 3-5 per mil higher in d15N values than their primary prey. A recent paper that includes a summary of previous collagen-to-collagen trophic enrichment factors found that a d15N TEF of 3.2 +- 1.8 was best at predicting diet (Krajcarz, Krajcarz, and Bocherens 2018). Additionally, in a presentation of the best practices for stable isotope mixing models Cheung & Szpak (2020) use a value of 4.0 +- 0.74 for the nitrogen TEF. In contrast to previous studies such as these, Chen and colleagues cite a paper by O’Connell et al. (2012) that studied d15N values in human red blood cells (not bone). From the blood data, the study then made a series of calculations to convert the blood data into collagen data, suggesting a value of about 6 per mil separated d15N from collagen and diet. This is notably very different than most other studies and from the accepted norm. Because a TEF of 6 per mil is nearly double what many scholars would consider to be an acceptable value for trophic enrichment (i.e. ~3), Chen and colleagues are essentially biasing the data towards a plant-focused diet instead of a diet that included significant herbivore meat, which indeed the zooarchaeological data from these two sites and from the broader region suggest.

The mean d15N value of the adults in the present study is 9.5 per mil while the mean d15N value of terrestrial mammals is 5.7, which means that the human-herbivore offset is 3.8 per mil. An offset of 3.8 is actually higher than the mean TEF suggested by Krajcarz et al. (2018) and within the range of the TEF suggested by Cheung & Szpak (2020). Using a more commonly accepted TEF would thus indicate that the humans from the SMP and WMP sites were primarily consuming large terrestrial mammals for their protein source. This is a very different conclusion than the one arrived at by Cheung et al. Because the entire argument of this paper relies on the interpretation of the nitrogen isotope data, their use of this exceptionally large TEF value needs further justification, or it should be changed to a lower, more commonly accepted value. Currently, I am unconvinced that these isotopic data imply low mean consumption. In fact, to me, they look like large herbivores were the dominant protein source.

Similarly, there are some issues with the use of the MixSIAR Bayesian mixing model. The authors do not mention what TEF and error range they use for the model. I infer that it was 6, but what was the error. What would the results be if the model was rerun with a TEF of 4 +-0.74? Also, one of the advantages of using a Bayesian stable isotope mixing model is the ability to incorporate prior knowledge into the model. Did the authors take advantage of the zooarchaeological and paleoethnobotanical data and use the prior function within the model? Or did they use all uninformed priors? (1, 1, 1, 1) Please state.

Additionally, I have few smaller suggestions:

Page 2, line 35: “For example, Rick [4]290 concluded that, “The settlement pattern [of the Junín region] and the faunal collections [from the site of Pachamachay] strongly support the hypothesis that vicuña, or similar camelids, were the major food source for puna [ecosystem] hunter-gatherers.” – This seems like an unnecessary quote that could be put into own words and summarized. In fact, it would be more impactful to mention why they claimed that; perhaps the high abundance of camelid remains found with the faunal collections of X sites led researchers, such as Rick {4} to conclude that…

Page 2, line 56: “The biases of previous researchers who tend to be males from a culture in which hunting is a distinctly masculine pursuit could furthermore inflate the hunting signal [23].” – Here it sounds strange to be discussing the biases of “previous researchers” using the present tense. I suggest changing to past tense.

Page 2, line 58” “It was ostensibly for this reason that ethnography famously revealed plant foods to play a prominent role in forager economies, in contrast to earlier models that emphasized hunting.” – This is too vague. What do you mean by ethnography? Some ethnographies demonstrate extreme levels of hunting (e.g. Arctic). Need to be more specific to make the point here.

Page 6, line 109. Why mentioning quality control after presenting the results? Seems more logical to assure the readers that the data are credible before you present them.

Page 8, first sentence of discussion section: How did the faunal data indicate that plants comprised the major component of early forager diets?

Reviewer #3: The authors present a paper based on an exciting question about the human diet in the early occupation of the Andes. The authors propose that their results refute the generally assumed idea that early hunter-gatherers based their diet on the exploitation of faunas, not plants or fish. For this, their research is based on the results of stable isotope analysis on human bone collagen (C and N) from two archaeological sites around 9K and 8K (Soro Mik’aya Patjxa and Wilamaya Patjxa). To interpret these results, they use stable isotope results in both modern and archaeological plants and animals (C and N).

Comments

*The Table “Human bone collagen isotopic results for Soro Mik’aya Patjxa and Wilamaya Patjxa Individuals” need to include %C, %N and C:N for each individual. It is essential to accept or reject the stable isotope value obtained. It is recommended to previously published values and other non-previously published ones. My question arise from the text where the authors mention, “As a quality control measure, we compare the resultant δ13C and δ15N values for three of the individuals—SMP9, SMP16, and WMP6—to values previously reported in radiocarbon analyses performed by The University of Arizona AMS laboratory and the KCCAMS facilty. We furthermore use C/N ratios in the radiocarbon samples to assess the extent of diagenetic alterations to bone collagen . ... We furthermore use C/N ratios in the radiocarbon samples to assess the extent of diagenetic alterations to bone collagen …”. This control measure, to my criteria, is not appropriated at all to publish stable isotope data today. Other aspect: how does the author consider/correct by breast-feeding in children individuals? Can you include it in the Table?

I recommend a figure/map indicating where the human remains are located and where the resources to interpret the human diet are localized. It is basic as far as any diet reconstruction is based mostly on the resource base data and discrimination/fraction factor used. The authors need to be confident the stable isotope value on resources are based on samples recorded close to or at least in a region with similar climate and ecology. Nitrogen is sensitive to climate, as example. Please give more details in Supplementary Table 1 about the palnts and faunas used to reconstruct the human diet (loacgtions (Latitude/longitude) altitude, modern/archaeological, %C, %N, C:N,lab code...

A central variable to reconstruct isotopic diet is the discrimination fraction values used to evaluate the proportion of resources. The authors mention, “..Raw δ13C and δ15N values are converted to dietary values using published fractionation rates to facilitate comparison with isotopic values of potential food resources. δ13Ccollagen values were converted to δ13Cdiet values using a -5‰ offset to account for fractionation during bone incorporation. δ15Ncollagen values are converted to δ15Ndiet values with an offset of -6‰ to account for trophic-level fractionation.” The autor needs to explain and justify this value. Other research proposes 3,5‰. Change in this value can change the author's interpretations. At the same time, the 13C fraction between faunas bone collagen and human bone collagen need be explicit (all corrected by 5‰? animals and plants?). I recommend the authors use a plot with raw data isotopic data first, then a second plot with the data corrected by fraction. I recommend the author discuss the fraction values used and why they reject others in more detail.

That is, the authors must justify the validity of the resources used and the fractionation values used to reconstruct human diets. Bernal et al. (Current Anthropology) show how strongly our interpretations of human diet reconstruction depend on these values.

The author indicates they used MixSiar to reconstruct the human diet with Bayesian statistics. Why not FRUIT? In the Supplementary material, I recommend details of the Mixiar (or Fruit) analysis. If the tuber should be the plant consumed, are the authors running Bayesian models with tubers or all C3 plants?

6. PLOS authors have the option to publish the peer review history of their article (what does this mean?). If published, this will include your full peer review and any attached files.

Reviewer #1: No

Reviewer #2: No

Reviewer #3: No

---

## [Author Response · Author response to Decision Letter 0]

27 Oct 2023

added as separate file per Editor instructions.

---

## [Decision Letter · Decision Letter 1]

4 Dec 2023

PONE-D-23-22494R1Stable isotope chemistry reveals plant-dominant diet among early foragers on the Andean Altiplano, 9.0–6.5 cal. kaPLOS ONE

Dear Dr. Haas,

Thank you for submitting your manuscript to PLOS ONE. After careful consideration, we feel that it has merit but does not fully meet PLOS ONE’s publication criteria as it currently stands. Therefore, we invite you to submit a revised version of the manuscript that addresses the points raised during the review process.

Dear authors, thanks for submitting the new version of this quite interesting manuscript, and thanks for addressing all of the editor and reviewers' suggestions/questions. I think this is a highly improved version and the last changes requested from reviewer 4 are truly minor. 

We look forward to receiving your revised manuscript.

Kind regards,

Miguel Delgado, PhD Division Anthropology, Faculty of Natural Sciences and Museum, National University of La Plata. 

Academic Editor

PLOS ONE

Journal Requirements:

Additional Editor Comments:

I want to share with the authors my opinion regarding the model used. Despite the authors having used various criteria to select the best model, the narrative that hunting was not common in the Archaic would be inconsistent with another viable model: d13C TEF of 5 and a d15N TEF of 4 (per Cheung and Szpak 2020) which would result in much lower plant contribution to the diet (75%). 

Reviewers' comments:

Reviewer's Responses to Questions

**Comments to the Author**

1. If the authors have adequately addressed your comments raised in a previous round of review and you feel that this manuscript is now acceptable for publication, you may indicate that here to bypass the “Comments to the Author” section, enter your conflict of interest statement in the “Confidential to Editor” section, and submit your "Accept" recommendation.

Reviewer #2: All comments have been addressed

Reviewer #4: (No Response)

2. Is the manuscript technically sound, and do the data support the conclusions?

Reviewer #2: Yes

Reviewer #4: Yes

3. Has the statistical analysis been performed appropriately and rigorously? 

Reviewer #2: Yes

Reviewer #4: Yes

4. Have the authors made all data underlying the findings in their manuscript fully available?

Reviewer #2: Yes

Reviewer #4: Yes

5. Is the manuscript presented in an intelligible fashion and written in standard English?

Reviewer #2: (No Response)

Reviewer #4: Yes

6. Review Comments to the Author

Reviewer #2: The manuscript is much improved from the original submission and I appreciate the extensive new modeling that the authors have done to address the issue of trophic discrimination factors and how they can lead to variable results. I am recommending the article be accepted.

Reviewer #4: In Tables 1 and 2, should only have one decimal place for the C and N isotope values. Same in lines 219-220.

In Table 4, the N isotope values should have one decimal place.

Otherwise, the authors have addressed all previous issues.

7. PLOS authors have the option to publish the peer review history of their article (what does this mean?). If published, this will include your full peer review and any attached files.

Reviewer #2: No

Reviewer #4: **Yes: **Robert H. Tykot

---

## [Author Response · Author response to Decision Letter 1]

6 Dec 2023

We have changed the significant figures on our isotope reporting per Reviewer 4's suggestion.

---

## [Editor Report · Decision Letter 2]

12 Dec 2023

Stable isotope chemistry reveals plant-dominant diet among early foragers on the Andean Altiplano, 9.0–6.5 cal. ka

PONE-D-23-22494R2

Dear Dr. Haas,

We’re pleased to inform you that your manuscript has been judged scientifically suitable for publication and will be formally accepted for publication once it meets all outstanding technical requirements.

Kind regards,

Miguel Delgado, PhD Division of Anthropology, Faculty of Natural Sciences and Museum, National University of La Plata.

Academic Editor

PLOS ONE
---

## [Editor Report · Acceptance letter]

18 Dec 2023

PONE-D-23-22494R2 

PLOS ONE

Dear Dr. Haas, 

I'm pleased to inform you that your manuscript has been deemed suitable for publication in PLOS ONE. Congratulations! Your manuscript is now being handed over to our production team.

Kind regards, 

on behalf of

Dr. Miguel Delgado 

Academic Editor

PLOS ONE